# VADv2: End-to-End Vectorized Autonomous Driving via Probabilistic Planning

**Bo Jiang**[1,*,◇], **Shaoyu Chen**[2,*], **Hao Gao**[1,◇], **Bencheng Liao**[1,◇],
**Qian Zhang**[2], **Wenyu Liu**[1], **Xinggang Wang**[1,⊠]

[1]Huazhong University of Science and Technology    [2]Horizon Robotics
{bjiang, ghao, bcliao, liuwy, xgwang}@hust.edu.cn
{shaoyu.chen, qian01.zhang}@horizon.auto

## Abstract

Learning a human-like driving policy from large-scale driving demonstrations is promising, but the uncertainty and non-deterministic nature of planning make it challenging. Existing learning-based planning methods follow a deterministic paradigm to directly regress the action, failing to cope with the uncertainty problem. In this work, we propose a probabilistic planning model for end-to-end autonomous driving, termed VADv2. We resort to a probabilistic field function to model the mapping from the action space to the probabilistic distribution. Since the planning action space is a high-dimensional continuous spatiotemporal space and hard to tackle, we first discretize the planning action space to a large planning vocabulary and then tokenize the planning vocabulary into planning tokens. Planning tokens interact with scene tokens and output the probabilistic distribution of action. Mass driving demonstrations are leveraged to supervise the distribution. VADv2 achieves state-of-the-art closed-loop performance on the CARLA Town05 benchmark, significantly outperforming existing methods, and also leads the recent Bench2Drive benchmark. We further provide comprehensive evaluations on NAVSIM and a large-scale 3DGS-based benchmark, demonstrating its effectiveness in real-world applications. Code is available at https://github.com/hustvl/VAD.

## 1    Introduction

End-to-end autonomous driving is an important research topic currently. Large-scale human driving demonstrations in real-world scenarios are readily available. It is very promising to derive a human-like driving policy for vehicle control from these extensive demonstrations. However, the uncertainty and non-deterministic nature of planning make it challenging to extract the driving knowledge from driving demonstrations. To illustrate such uncertainty, two scenarios are presented in Figure 1 and explained as follows. 1) Following another vehicle: The human driver exhibits various reasonable driving maneuvers, such as maintaining the current lane or changing lanes to overtake. 2) Interaction with an oncoming vehicle: The human driver faces two potential driving maneuvers, i.e., yielding or overtaking. From a statistical perspective, the actions (including timing and speed) are highly stochastic, influenced by numerous latent factors that cannot be accurately modeled.

Existing learning-based planning methods (Jiang et al., 2023; Hu et al., 2022c; Jia et al., 2023b; Prakash et al., 2021b; Hu et al., 2022a; Zhang et al., 2021) follow a deterministic paradigm to directly regress the action. The regression target is the future trajectory in (Jiang et al., 2023; Hu et al., 2022c; Jia et al., 2023b; Prakash et al., 2021b) and control signal (acceleration and steering) in (Hu et al., 2022a; Zhang et al., 2021). Such a paradigm assumes there exists a deterministic relation between the driving scene and action, which is not the case. The variance of human driving behavior causes the ambiguity of the regression target. Especially when the feasible solution space is non-convex, *i.e.*, there exist multiple feasible solutions (see Figure 1), deterministic modeling may fail to properly handle such cases and produce an intermediate action, resulting in potential safety risks.

---

*Equal contribution. ◇ Work done during internship at Horizon Robotics. ⊠ Corresponding author.

In this work, we propose probabilistic planning to cope with the uncertainty of planning. We model the planning policy as a scene-conditioned non-stationary stochastic process, formulated as $p(\boldsymbol{a}|\boldsymbol{o})$, where $\boldsymbol{o}$ is the historical and current observations of the driving scene, and $\boldsymbol{a}$ is a candidate planning action. Compared with deterministic planning, probabilistic planning is more robust against uncertainty in planning and able to model non-convex feasible solution space, and thus achieves more accurate and safer planning.

We resort to a probabilistic field function to model the mapping from the planning action space to the probabilistic distribution. Since the action space is a high-dimensional continuous spatiotemporal space and hard to tackle, we first discretize the planning action space to a large planning vocabulary and then tokenize the planning vocabulary into planning tokens. Planning tokens interact with scene tokens and output the probabilistic distribution of action. Mass driving demonstrations are leveraged to supervise the distribution.

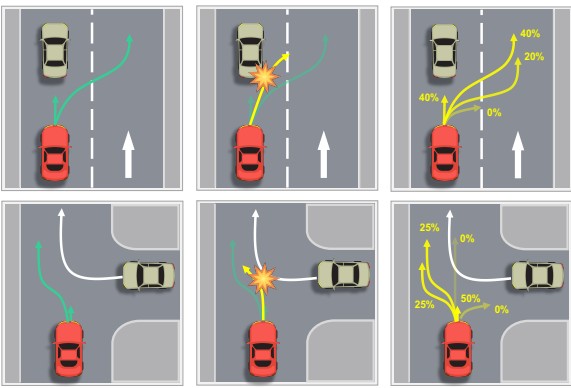

**Driving Demonstrations**  **Deterministic Planning**  **Probabilistic Planning**

Figure 1: Uncertainty exists in planning. There doesn't exist a deterministic relation between driving scene and action. The deterministic planning fails to model such uncertainty especially when the feasible solution space is non-convex. VADv2 is based on probabilistic planning and learns scene-conditioned probabilistic distribution of action from large-scale driving demonstrations.

Probabilistic planning has two other advantages. First, unlike deterministic planning which has to regress the optimal action based on scene information, probabilistic planning models the correlation between each action and the driving scene. It just ranks different actions and samples a high-scoring one. Such modeling is much simpler. Besides, probabilistic planning is flexible in the inference stage. It outputs multi-mode planning results and is easy to combine with rule-based and optimization-based planning methods. We can flexibly add other candidate planning actions to the planning vocabulary and evaluate them because we model the distribution over the whole action space.

Based on the probabilistic planning, we present VADv2, an end-to-end driving model, which takes surround-view image sequence as input in a streaming manner, tokenizes sensor data and planning action space, outputs the probabilistic distribution of action, and samples one action to control the vehicle. Using only camera sensors, VADv2 achieves state-of-the-art closed-loop performance on the CARLA Town05 benchmark, significantly outperforming existing methods, and also leads the recent Bench2Drive benchmark. It further delivers strong planning performance on NAVSIM and our 3DGS-based benchmark. VADv2 runs stably in a fully end-to-end manner, even without a rule-based wrapper as a post-processing step to avoid infraction.

Our contributions are summarized as follows:

- We propose probabilistic planning to cope with the uncertainty and non-deterministic nature of planning. We design a probabilistic field to map from the action space to the probabilistic distribution and learn the distribution of action from large-scale driving demonstrations.
- Based on the probabilistic planning, we present VADv2, an end-to-end driving model, which tokenizes sensor data and planning action space for interaction, outputs the probabilistic distribution of action, and samples one action to control the vehicle.
- VADv2 achieves state-of-the-art planning performance in both closed- and open-loop settings across multiple benchmarks. Abundant closed-loop simulations and real-world deployment results validate its effectiveness and stability in vehicle control.

## 2  RELATED WORK

**Perception.** Perception is the first step in achieving autonomous driving, and a unified representation of driving scenes is beneficial for easy integration into downstream tasks. Bird's Eye View (BEV)

representation has become a common strategy in recent years, enabling effective scene feature encoding and multimodal data fusion. LSS (Philion & Fidler, 2020) is a pioneering work that achieves the perspective view to BEV transformation by explicitly predicting depth for image pixels. BEVFormer (Li et al., 2022c), on the other hand, avoids explicit depth prediction by designing spatial and temporal attention mechanisms. Subsequent works (Li et al., 2022b; Wang et al., 2023a) continuously optimize temporal modeling and BEV transformation strategies. In terms of vectorized mapping, HDMapNet (Li et al., 2022a) converts lane segmentation into vector maps through post-processing. VectorMapNet (Liu et al., 2022) predicts vector map elements in an autoregressive manner. MapTR (Liao et al., 2022; 2023b) introduces permutation equivalence and hierarchical matching strategies, significantly improving mapping performance. LaneGAP (Liao et al., 2023a) introduces path-wise modeling for lane graphs.

**Motion Prediction.** Motion prediction aims to forecast future trajectories of other traffic participants, assisting the ego vehicle in making informed planning decisions. Traditional motion prediction methods utilizes input such as historical trajectories and high-definition maps to predict future trajectories (Gao et al., 2020; Liu et al., 2021). However, recent end-to-end methods (Gu et al., 2022; Jiang et al., 2022) perform perception and motion prediction jointly. Some works (Hu et al., 2021; Zhang et al., 2022) represent future motion as dense occupancy and flow fields, while others (Gu et al., 2022; Jiang et al., 2022) predict agent-level multi-modal trajectories. Another line of work reformulate trajectory prediction as a classification problem rather than a regression task. Trajeglish (Philion et al., 2023) introduces K-disk sampling to construct a compact one-step motion vocabulary, achieving lower discretization error compared to k-means. MotionLM (Seff et al., 2023) factorizes each single-step action into longitudinal and lateral components and applies axis-aligned uniform quantization. While such single-step modeling enables a compact representation and a small vocabulary, iterative rollout can lead to error accumulation and may produce trajectories that violate physical constraints. In contrast, each action token in VADv2 represents a complete trajectory, ensuring physically feasible motion primitives and enabling one-shot planning without error accumulation.

**Planning.** Learning-based planning has shown great potential recently due to its data-driven nature and impressive performance with increasing amounts of data. Early attempts (Codevilla et al., 2019; Prakash et al., 2021a) use a completely black-box spirit, where sensor data is directly used to predict control signals. However, this strategy lacks interpretability and is difficult to optimize. In addition, there are numerous studies combining reinforcement learning and planning (Zhang et al., 2021; Gao et al., 2025) by autonomously exploring driving behavior in closed-loop simulation environments. Imitation learning (Chekroun et al., 2021; Hu et al., 2022b; Ma et al., 2025) is another research direction, where models learn expert driving behavior to achieve good planning performance and develop a driving style close to that of humans.

UniAD (Hu et al., 2022c) integrates multiple perception and prediction tasks to enhance planning performance. VAD (Jiang et al., 2023) explores the potential of vectorized scene representation for planning and getting rid of dense maps. Diffusion Planner (Zheng et al., 2025) jointly predicts ego and other agents' motions via iterative diffusion, but it relies on ground-truth perception and HD maps. DiffusionDrive (Liao et al., 2025) accelerates the denoising process through truncated diffusion; yet the limited and predefined trajectory anchors may constrain generation quality, and increasing the number of anchors introduces additional computational cost. GoalFlow (Xing et al., 2025) decomposes planning into goal-point selection followed by trajectory generation via flow matching, but relying on a single goal point may affect trajectory diversity, and its hand-crafted trajectory selection rules limit generality and scalability.

**Large Language Model in Autonomous Driving.** Recent researches explore the combination of LLMs and autonomous driving (Sha et al., 2023; Xu et al., 2023). One line of work utilizes LLMs for driving scene understanding and evaluation through question-answering (Chen et al., 2023; Sima et al., 2024). Another approach goes a step further by directly utilizing LLMs for planning (Wang et al., 2023b; 2024). However, current LLM-based planning approaches inevitably suffer from limited inference speed, which constrains their practicality for real-time deployment in autonomous driving applications. VADv2 draws inspiration from GPT (Achiam et al., 2023) to cope with the uncertainty problem, which also exists in language modeling. Given a specific context, the next word is non-deterministic, LLM learns the context-conditioned probabilistic distribution of the next word from a large-scale corpus, and samples one word from the distribution. Inspired by LLM, VADv2 models

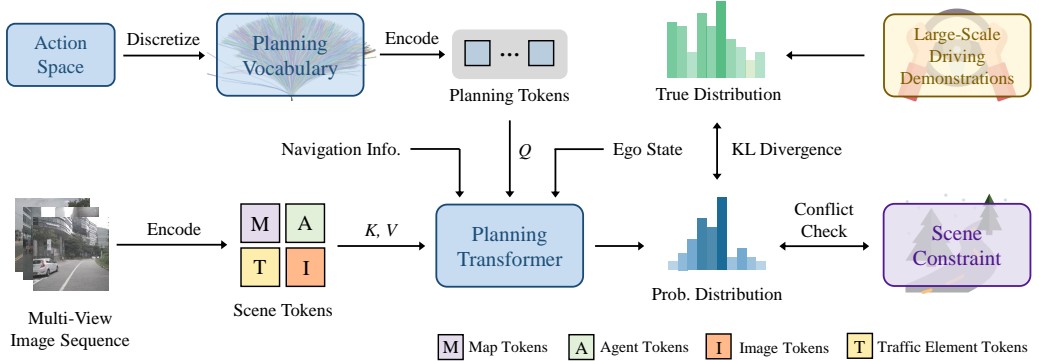

Figure 2: **Overall architecture of VADv2.** VADv2 takes multi-view image sequences as input in a streaming manner, tokenizes sensor data and planning action space, outputs the probabilistic distribution of action, and samples one action to control the vehicle. Large-scale driving demonstrations and scene constraints are used to supervise the predicted distribution.

the planning policy as a scene-conditioned nonstationary stochastic process. VADv2 discretizes the action space to generate a planning vocabulary, approximates the probabilistic distribution based on large-scale driving demonstrations, and samples one action from the distribution at each time step to control the vehicle.

# 3 VADv2

The overall framework of VADv2 is depicted in Figure 2. VADv2 takes multi-view image sequences as input in a streaming manner, transforms sensor data into scene token embeddings, outputs the probabilistic distribution of action, and samples one action to control the vehicle. Large-scale driving demonstrations and scene constraints are used to supervise the predicted distribution.

## 3.1 SCENE ENCODER

VADv2 uses a scene encoder to transform the sensor data into instance-level token embeddings $E_{\text{scene}} \in \mathbb{R}^{M \times D}$ to explicitly extract high-level information, where $M$ is the number of scene tokens and $D$ is the feature dimension. Concretely, $E_{\text{scene}}$ includes four kinds of scene tokens: map tokens, agent tokens, traffic element tokens, and image tokens.

**BEV Encoder.** A BEV encoder (Li et al., 2022c) is first employed to transform the multi-view image features from perspective view into Bird's Eye View, producing a feature map in the BEV space. This feature map serves as the basis for learning instance-level map and agent features.

**Map Tokens.** A set of map tokens (Liao et al., 2023b) is introduced to learn vectorized map elements of the driving scene from the BEV feature map, including lane centerlines, lane dividers, road boundaries, and pedestrian crossings.

**Agent Tokens.** In addition, a group of agent tokens (Jiang et al., 2023) is adopted to predict motion information of traffic participants, including location, orientation, size, speed, and future trajectories.

**Traffic Element Tokens.** Traffic signals also play a vital role in planning. In CARLA, we consider two types of traffic signals: traffic lights and stop signs. Front-view image features extracted from the backbone are further encoded using an MLP into traffic element tokens, which are then used to predict the states of the traffic signals.

**Image Tokens.** Apart from the above instance-level tokens, front-view image features are also used as image tokens. These image tokens provide denser scene features for planning, capturing rich information that complements the instance-level tokens.

Map tokens, agent tokens, and traffic element tokens are supervised with corresponding supervision signals to ensure they explicitly encode corresponding high-level information. Besides, navigation information and ego state are also encoded into embeddings ($E_{\text{navi}}$, $E_{\text{state}}$) with an MLP. In summary, the Scene Encoder transforms the sparse sensor data into more compact high-level scene features ($E_{\text{scene}}$, $E_{\text{navi}}$, $E_{\text{state}}$), which serve as the foundation for the following planning module.

## 3.2 PROBABILISTIC PLANNING

We propose probabilistic planning to cope with the uncertainty of planning. We model the planning policy as a scene-conditioned nonstationary stochastic process, formulated as $p(\boldsymbol{a}|\boldsymbol{o})$, where $\boldsymbol{o}$ is the observed scene information and $\boldsymbol{a}$ is the action, represented as a waypoint sequence of future planning trajectory,

$$\boldsymbol{o} = (E_{\text{scene}},\ E_{\text{navi}},\ E_{\text{state}}),\ \ \boldsymbol{a} = (x_1,\ y_1,\ x_2,\ y_2, \ldots,\ x_T,\ y_T). \tag{1}$$

$T$ is the number of waypoints. Each waypoint $(x_i, y_i)$ corresponds to a future timestamp $t_i$.

We approximate the probabilistic distribution of the planning action space based on large-scale driving demonstrations, and sample one action from the distribution at each time step to control the vehicle.

The planning action space is a high-dimensional continuous spatiotemporal space and hard to tackle. Thus, we discretize the planning action space to a large planning vocabulary $\mathcal{V} = \{\boldsymbol{a}^i\}^N$, where $N$ is the vocabulary size. Specifically, we collect all the planning actions in driving demonstrations as the planning action set $\mathcal{S}$ and adopt the furthest trajectory sampling to select $N$ representative actions to serve as the planning vocabulary. The vocabulary sampling algorithm is presented in Algorithm 1. Each planning action in $\mathcal{V}$ is sampled from driving demonstrations and thus naturally satisfies the kinematic constraints of the ego vehicle, which means that when the action is converted into control signals (steer, throttle, and brake), the control signal values do not exceed the feasible range. By default, $N$ is set to 4096.

The probability $p(\boldsymbol{a})$ is assumed to be continuous with respect to $\boldsymbol{a}$ and insensitive to small perturbations in $\boldsymbol{a}$, *i.e.*, $\lim_{\Delta\boldsymbol{a}\to 0}(p(\boldsymbol{a}) - p(\boldsymbol{a} + \Delta\boldsymbol{a})) = 0$. Inspired by NeRF (Mildenhall et al., 2020), which models the continuous radiance field over the 5D space, we resort to a probabilistic field to model the continuous mapping from the action space to the probabilistic distribution.

---

**Algorithm 1:** Planning vocabulary sampling.

**Input:** Planning action set $\mathcal{S}$, Planning vocabulary size $N$
**Output:** Planning vocabulary $\mathcal{V}$
Initialization: $\mathcal{V} \leftarrow \emptyset$
**for** $i = 1$ *to* $N-1$ **do**
  **if** $\mathcal{V} == \emptyset$ **then**
    random select a action $\boldsymbol{a}$ in $\mathcal{S}$
    $\mathcal{V} \leftarrow \mathcal{V} \cup \{\boldsymbol{a}\}$ add action to planning vocabulary
    $\mathcal{S} \leftarrow \mathcal{S} \setminus \{\boldsymbol{a}\}$ remove action from planning action set
  **end**
  $max\_dis = 0$
  **for** trajectory $\boldsymbol{a}$ *in* $\mathcal{S}$ **do**
    $dis = \textbf{calculate\_distance}(\boldsymbol{a}, \mathcal{V})$
    **if** $dis > max\_dis$ **then**
      $max\_dis = dis$
      $\hat{\boldsymbol{a}} = \boldsymbol{a}$ update the currently furthest trajectory
    **end**
  **end**
  $\mathcal{V} \leftarrow \mathcal{V} \cup \{\hat{\boldsymbol{a}}\}$
**end**
Return: $\mathcal{V}$

---

**calculate_distance** will calculate the distance between the endpoint of action $\boldsymbol{a}$ and the endpoints of all actions in set $\mathcal{V}$. It will return the minimum value among these distances as the result.

Concretely, we first encode each action (trajectory waypoint) into a high-dimensional planning token embedding $E(\boldsymbol{a})$,

$$E(\boldsymbol{a}) = \big(\Gamma(x_i),\ \Gamma(y_i)\big)_{i=1}^T,\ \Gamma(\text{pos}) = \big(\gamma(\text{pos},\ j)\big)_{j=0}^{L-1},$$
$$\gamma(\text{pos},\ j) = \big(\cos(\text{pos}/10000^{2\pi j/L}),\ \sin(\text{pos}/10000^{2\pi j/L})\big). \tag{2}$$

$\Gamma$ is an encoding function that maps each coordinate from $\mathbb{R}$ into a high dimensional embedding space $\mathbb{R}^{2L}$, and is applied separately to each of the coordinate values of trajectory $\boldsymbol{a}$. pos denotes the

position (referring to the $x$ or $y$ coordinate of waypoint). We use these functions to map continuous coordinates into a higher dimensional space to better approximate a higher frequency field function.

Then, a cascaded Transformer decoder $\phi$ interacts with scene information $E_{\text{scene}}$ and, combined with navigation $E_{\text{navi}}$ and ego state $E_{\text{state}}$, predicts the probability of each action,

$$p(\boldsymbol{a}) = \sigma(\text{MLP}(\phi(E(\boldsymbol{a}),\ E_{\text{scene}}) + E_{\text{navi}} + E_{\text{state}})). \tag{3}$$

$\sigma$ is the sigmoid function. In the Transformer decoder $\phi$, $E(\boldsymbol{a})$ serves as query, and $E_{\text{scene}}$ serves as key and value. $E(\boldsymbol{a})$, $E_{\text{navi}}$, $E_{\text{state}}$, and the output of MLP are with the same dimension ($1 \times D$).

## 3.3 TRAINING

We train VADv2 with three kinds of supervision, distribution loss, conflict loss, and scene token loss.

**Distribution Loss.** We learn the probabilistic distribution from large-scale driving demonstrations. KL divergence is used to minimize the difference between the predicted distribution and the distribution of the data:

$$\mathcal{L}_{\text{distribution}} = D_{\text{KL}}(p_{\text{data}}||p_{\text{pred}}) = \sum_{\boldsymbol{a} \in \mathcal{V}} p_{\text{data}}(\boldsymbol{a}) \cdot \log \frac{p_{\text{data}}(\boldsymbol{a})}{p_{\text{pred}}(\boldsymbol{a})}, \tag{4}$$

$p_{\text{data}}(\boldsymbol{a})$ is estimated through occurrence frequency in demonstrations. Since $p_{\text{data}}(\boldsymbol{a})$ is fixed, $p_{\text{data}}(\boldsymbol{a}) \cdot \log p_{\text{data}}(\boldsymbol{a})$ is a constant and can be omitted. Therefore, minimizing KL divergence is equivalent to optimizing the cross-entropy loss:

$$\mathcal{L}_{\text{distribution}} = -\sum_{\boldsymbol{a} \in \mathcal{V}} p_{\text{data}}(\boldsymbol{a}) \cdot \log p_{\text{pred}}(\boldsymbol{a}). \tag{5}$$

For each frame in the demonstrations, we select the action from the planning vocabulary that has the lowest L2 distance to the ground-truth action. This best-matched action is assigned a label of 1, and all other actions are assigned 0. Over all the frames, the occurrence frequency $p_{\text{data}}(\boldsymbol{a})$ of one action $\boldsymbol{a}$ is then estimated by counting how often each action is the best match, normalized by the total number of frames. This modeling is analogous to the standard formulation used in large language models, where the ground-truth token is labeled as 1 and others as 0, and cross-entropy loss is used to minimize the divergence between the predicted distribution and the empirical distribution.

**Conflict Loss.** Driving scene constraints are used to help model learn important driving priors and regularize the predicted action distribution. Specifically, if an action in the planning vocabulary conflicts with other agents' ground truth future motion or road boundaries, it is treated as a negative sample, and a corresponding loss is applied to reduce its probability,

$$\mathcal{L}_{\text{conflict}} = \sum_{\boldsymbol{a} \in \mathcal{V}} \mathbb{1}_{\text{conflit}}(\boldsymbol{a}) \cdot \log p_{\text{pred}}(\boldsymbol{a}). \tag{6}$$

$\mathbb{1}_{\text{conflit}}(\boldsymbol{a})$ is the indicator function, whose value is 1 if conflict happens to $\boldsymbol{a}$, otherwise is 0.

**Scene Token Loss.** Map, agent, and traffic-element tokens are supervised with corresponding supervision signals to ensure they explicitly encode the relevant high-level information.

The loss of map tokens is the same with MapTRv2 (Liao et al., 2023b). $l_1$ loss is adopted to calculate the regression loss between the predicted map points and the ground truth map points. Focal loss is used as the map classification loss.

The loss of agent tokens is composed of the detection loss and the motion prediction loss (Jiang et al., 2023). $l_1$ loss is used as the regression loss to predict agent attributes (location, orientation, size, *etc.*), and focal loss to predict agent classes. For each agent who has matched with a ground truth agent, we predict $K$ future trajectories and use the trajectory that has the minimum final displacement error (minFDE) as a representative prediction. Then we calculate $l_1$ loss between this representative

Table 1: Closed-loop evaluation on the Town05 Long benchmark.

| Method | Reference | Modality | DS ↑ | RC ↑ | IS ↑ |
|---|---|---|---|---|---|
| Transfuser (Prakash et al., 2021b) | TPAMI'22 | C+L | 31.0 | 47.5 | 0.77 |
| ThinkTwice (Jia et al., 2023b) | CVPR'23 | C+L | 70.9 | 95.5 | 0.75 |
| DriveAdapter+TCP (Jia et al., 2023a) | ICCV'23 | C+L | 71.9 | 97.3 | 0.74 |
| DriveMLM (Wang et al., 2023b) | arXiv'23 | C+L | 76.1 | 98.1 | 0.78 |
| Roach (Zhang et al., 2021) | ICCV'21 | C | 41.6 | 96.4 | 0.43 |
| ST-P3 (Hu et al., 2022b) | ECCV'22 | C | 11.5 | 83.2 | - |
| MILE (Hu et al., 2022a) | NeurIPS'22 | C | 61.1 | 97.4 | 0.63 |
| Interfuser (Shao et al., 2023) | CoRL'22 | C | 68.3 | 95.0 | - |
| VAD (Jiang et al., 2023) | ICCV'23 | C | 30.3 | 75.2 | - |
| Rao et al. (2024) | TIV'24 | C | 74.9 | 94.6 | 0.77 |
| DriveCoT (Wang et al., 2024) | arXiv'24 | C | 73.6 | 96.8 | 0.76 |
| LeapVAD (Ma et al., 2025) | arXiv'25 | C | 73.7 | 95.7 | 0.78 |
| VADv2 | Ours | C | **85.1** | **98.4** | **0.87** |

Table 2: End-to-end planning results on the NAVSIM `navtest` split with closed-loop metrics.

| Method | Reference | Modality | NC ↑ | DAC ↑ | TTC ↑ | Comf ↑ | EP ↑ | PDMS ↑ |
|---|---|---|---|---|---|---|---|---|
| UniAD (Hu et al., 2022c) | CVPR'23 | C | 97.8 | 91.9 | 92.9 | 100 | 78.8 | 83.4 |
| Transfuser (Prakash et al., 2021b) | PAMI'23 | C+L | 97.7 | 92.8 | 92.8 | 100 | 79.2 | 84.0 |
| PARA-Drive (Weng et al., 2024) | CVPR'24 | C | 97.9 | 92.4 | 93.0 | 99.8 | 79.3 | 84.0 |
| GoalFlow Xing et al. (2025) | CVPR'25 | C+L | 98.3 | 93.8 | 94.3 | 100 | 79.8 | 85.7 |
| Hydra-MDP++ Li et al. (2025a) | arXiv'25 | C | 97.6 | 96.0 | 93.1 | 100 | 80.4 | 86.6 |
| DiffusionDrive (Liao et al., 2025) | CVPR'25 | C+L | 98.2 | 96.2 | 94.7 | 100 | 82.2 | 88.1 |
| WoTE Li et al. (2025b) | ICCV'25 | C+L | **98.5** | 96.8 | 94.9 | 99.9 | 81.9 | 88.3 |
| Hydra-NeXt Li et al. (2025c) | arXiv'25 | C+L | 98.1 | **97.7** | 94.6 | 100 | 81.8 | 88.6 |
| VADv2 | Ours | C | 98.3 | 97.4 | **95.7** | **100** | **82.3** | **89.3** |

trajectory and the ground truth trajectory as the motion regression loss. Besides, focal loss is adopted as the multi-modal motion classification loss.

Traffic element tokens consist of two parts: the traffic light token and the stop sign token. On one hand, we send the traffic light token to an MLP to predict the state of the traffic light (yellow, red, and green) and whether the traffic light affects the ego vehicle. On the other hand, the stop sign token is also sent to an MLP to predict the overlap between the stop sign area and the ego vehicle. Focal loss is used to supervise these predictions. The final loss can be denoted as:

$$\mathcal{L} = \mathcal{L}_{\text{distribution}} + \mathcal{L}_{\text{conflict}} + \mathcal{L}_{\text{token}}. \tag{7}$$

## 3.4 INFERENCE

In closed-loop inference, it's flexible to get the driving policy from the distribution. Intuitively, we sample the action with the highest probability at each time step, and use the PID controller to convert the selected action to control signals (steer, throttle, and brake).

In real-world applications, there are more robust strategies to make full use of the probabilistic distribution. A good practice is, sampling top-K actions as proposals, and adopting a rule-based wrapper for filtering proposals and an optimization-based post-solver for fine-grained trajectory refinement. Besides, the probability of the action reflects how confident the end-to-end model is, and can be regarded as the judgment condition to switch between conventional rule-based planning and control and learning-based planning and control.

## 4 EXPERIMENTS

### 4.1 EXPERIMENTAL SETTINGS

**CARLA Benchmark.** We first use CARLA (Dosovitskiy et al., 2017) simulator to evaluate the performance of VADv2. We conduct closed-loop evaluation on the widely adopted Town05 and

Table 3: End-to-end planning results on the NAVSIMv2 benchmark with Extended Metrics.

| Method | NC ↑ | DAC ↑ | DDC ↑ | TL ↑ | EP ↑ | TTC ↑ | LK ↑ | HC ↑ | EC ↑ | EPDMS ↑ |
|---|---|---|---|---|---|---|---|---|---|---|
| Transfuser | 96.9 | 89.9 | 97.8 | 99.7 | 87.1 | 95.4 | 92.7 | 98.3 | 87.2 | 76.7 |
| HydraMDP++ | 97.2 | 97.5 | 99.4 | 99.6 | 83.1 | 96.5 | 94.4 | 98.2 | 70.9 | 81.4 |
| PRIX | 98.0 | 95.6 | **99.5** | 99.8 | **87.4** | **97.2** | **97.1** | 98.3 | 87.6 | 84.2 |
| VADv2 | **98.0** | **98.3** | 99.4 | **99.8** | 87.1 | 96.8 | 95.2 | 98.3 | **88.1** | **85.8** |

Table 4: Closed-loop quantitative comparisons with other methods on the 3DGS-based benchmark.

| Method | Reference | CR↓ | DCR↓ | SCR↓ | DR↓ | PDR↓ | HDR↓ | ADD↓ |
|---|---|---|---|---|---|---|---|---|
| TransFuser | TPAMI 22 | 0.320 | 0.273 | 0.047 | **0.235** | 0.188 | **0.047** | **0.263** |
| VAD | ICCV 23 | 0.335 | 0.273 | 0.062 | 0.314 | 0.255 | 0.059 | 0.304 |
| GenAD | ECCV 24 | 0.341 | 0.299 | 0.042 | 0.291 | 0.160 | 0.131 | 0.265 |
| VADv2 | Ours | **0.270** | **0.240** | **0.030** | 0.243 | **0.139** | 0.104 | 0.273 |

Bench2Drive Jia et al. (2024) benchmarks. Specifically, each benchmark contains several pre-defined driving routes. The simulation and control frequency for closed-loop inference is 10 Hz. VADv2 takes a multi-view image sequence as input in a streaming manner and plans a 3-second future trajectory. The trajectory consists of 6 waypoints (*i.e.*, $T = 6$). The time interval between two adjacent waypoints is 0.5s. The default feature dimension $D$ for VADv2 is set to 256. All experiments are conducted based on 16 NVIDIA 4090 GPUs.

**CARLA Data.** As for generating the driving demonstration data for the Town05 benchmark, we use the official autonomous agent of CARLA to collect training data by randomly generating driving routes in Town03, Town04, Town06, Town07, and Town10. We collect approximately 3 million clips for training. For each clip, we save 6-camera surround-view image sequences at 10Hz for the past 1.6 seconds, along with information on traffic signals, traffic participants, and the state of the ego vehicle. Additionally, we obtain the vectorized maps for training the online mapping module by preprocessing the OpenStreetMap (Haklay & Weber, 2008) format maps provided by CARLA. The maps are provided only as ground truth during training, and VADv2 does not use high-definition maps for evaluation.

**NAVSIM and 3DGS-based Benchmarks.** To further validate generalization ability in real-world scenarios, we also evaluate VADv2 on the NAVSIM, NAVSIMv2 (Dauner et al., 2024) and a large-scale 3DGS-based (Kerbl et al., 2023) benchmarks. we collect 2000 hours of real-world human driving demonstrations for training, and utilizing 337 reconstructed 3D Gaussian Splatting (3DGS) environments for closed-loop evaluation. Each environment features an 8s scenario capturing interactions in dense traffic with potential collision risks, providing a representative segment of real-world driving behaviors and multi-agent interactions.

The photorealistic 3DGS reconstruction enables accurate agent trajectory modeling and dynamic environment rendering, providing a testbed that closely mirrors real-world driving conditions. More details of the 3DGS-based benchmark can be found in Appendix A due to page limits. We also deploy VADv2 on real-world vehicles. The results are presented in the supplementary material.

## 4.2 METRICS

On the CARLA benchmark, we employ its official closed-loop metrics, Route Completion (RC), Infraction Score (IS) and Driving Score (DS). DS is the product between RC and IS, which serves as the main metric. In benchmark evaluation, most works adopt a rule-based wrapper to reduce the infraction. For a fair comparison, we follow the common practice of adopting a rule-based wrapper over the learning-based policy, which is similar to Transfuser (Prakash et al., 2021b). We also conduct open-loop evaluation using the L2 displacement error and collision rate. In most ablations, we adopt open-loop metrics by default because they are faster to evaluate and more stable. We use the official autonomous agent of CARLA to generate the validation set on the Town05 Long benchmark for open-loop evaluation, and the results are averaged over all validation samples.

Table 5: Closed-loop results on the Bench2Drive.

| Method | DS ↑ | SR ↑ | Effi ↑ | Comf ↑ |
|---|---|---|---|---|
| SparseDrive | 44.54 | 16.71 | 170.21 | 48.63 |
| MomAD | 47.91 | 18.11 | 174.91 | **51.20** |
| DriveTransformer | 63.46 | 35.01 | 100.64 | 20.78 |
| ETA | 74.33 | 48.33 | **186.04** | 25.77 |
| VADv2 | **76.15** | **50.46** | 178.24 | 37.81 |

Table 6: Ablation on the multi-modal outputs.

| Traj. | NC ↑ | DAC ↑ | TTC ↑ | Comf ↑ | EP ↑ | PDMS ↑ |
|---|---|---|---|---|---|---|
| Top1 | **98.3** | **97.4** | **95.7** | 100 | **82.3** | **89.3** |
| Top2 | 98.2 | 97.3 | 94.8 | 100 | 82.2 | 89.1 |
| Top3 | 98.0 | 96.7 | 94.6 | 99.8 | 82.0 | 88.3 |
| Top4 | 97.8 | 96.2 | 93.2 | 99.9 | 81.7 | 87.6 |
| Top5 | 97.4 | 96.1 | 93.5 | 99.7 | 81.4 | 87.5 |

Table 7: **Ablations on design choices.** "Dist.": Distribution; "Traf. Token": Traffic Element Token.

| ID | Dist. Loss | Conflict Loss | Agent Token | Map Token | Traf. Token | Image Token | L2 (m) ↓ | | | Collision (%) ↓ | | |
|---|---|---|---|---|---|---|---|---|---|---|---|---|
| | | | | | | | 1s | 2s | 3s | 1s | 2s | 3s |
| 1 | | ✓ | ✓ | ✓ | ✓ | ✓ | 1.415 | 2.310 | 3.153 | 0.698 | 0.755 | 0.746 |
| 2 | ✓ | | ✓ | ✓ | ✓ | ✓ | 0.086 | 0.173 | 0.291 | 0.000 | 0.012 | 0.039 |
| 3 | ✓ | ✓ | | ✓ | ✓ | ✓ | 0.089 | 0.190 | 0.327 | 0.015 | 0.047 | 0.085 |
| 4 | ✓ | ✓ | ✓ | | ✓ | ✓ | 0.086 | 0.191 | 0.332 | 0.005 | 0.034 | 0.070 |
| 5 | ✓ | ✓ | ✓ | ✓ | | ✓ | 0.082 | 0.171 | 0.295 | 0.000 | 0.017 | 0.051 |
| 6 | ✓ | ✓ | ✓ | ✓ | ✓ | | 0.083 | 0.170 | 0.293 | 0.000 | 0.010 | 0.039 |
| 7 | ✓ | ✓ | ✓ | ✓ | ✓ | ✓ | 0.082 | 0.169 | 0.290 | 0.000 | 0.010 | 0.039 |

On the NAVSIM benchmark, its official closed-loop metrics such as PDMS are adopted. On the 3DGS-based benchmark, we use safety-critical metrics grounded in real-world driving analytics: Collision Ratio (CR, sum of Dynamic and Static Collision Ratios) evaluates interaction safety in dense traffic, while Deviation Ratio (DR, combining Positional and Heading Deviation Ratios) and Average Deviation Distance (ADD) jointly assess trajectory consistency with expert human demonstrations.

### 4.3 COMPARISONS WITH STATE-OF-THE-ART METHODS

On the Town05 Long benchmark in Table 1, VADv2 achieves a DS of 85.1, a RC of 98.4, and an IS of 0.87. Relative to the former state-of-the-art method DriveMLM (Wang et al., 2023b), VADv2 achieves a higher RC while significantly improving DS by 9.0. It is worth noting that VADv2 only utilizes cameras as perception input, whereas DriveMLM uses both cameras and LiDAR. Furthermore, compared to the previous best camera-based method (Rao et al., 2024), VADv2 demonstrates even greater advantages, with a remarkable increase in DS of up to 10.2. On the Bench2Drive benchmark (Table 5), VADv2 also achieves the highest Drive Score of 76.15.

Besides the CARLA-based benchmarks, VADv2 also achieves state-of-the-art planning performance on the NAVSIM and NAVSIMv2 benchmarks, as shown in Table 2 and Table 3. Additionally, Table 4 presents the results on the 3DGS-based benchmark. VADv2 reduces the Collision Ratio to 0.270, a 15.6% improvement over TransFuser (0.320), while maintaining a competitive Deviation Ratio of 0.243 that approaches the best reported performance. These results demonstrate robust safety in real-world dynamic interactions.

### 4.4 ABLATION STUDY

**Multimodal Planning Performance.** We further evaluate the multi-modal planning performance of VADv2. Beyond the highest-probability trajectory, we consider other candidates within the top-5 set, as shown in Table 6. Results indicate that the top-1 trajectory achieves the best overall performance, and the performance of other candidates is basically comparable to the top-1, demonstrating that VADv2 can generate high-quality multi-modal outputs.

**Key Modules.** Table 7 shows the ablation experiments of the key modules in VADv2. 50k clips of driving demonstrations are used in training. The model performs poorly in terms of planning accuracy without the supervision of expert driving behavior provided by the Distribution Loss (ID 1). The Conflict Loss provides critical prior information about driving; therefore, removing it (ID 2) also affects the model's planning accuracy. Scene tokens encode important scene elements into high-dimensional features, and the planning tokens interact with the scene tokens to learn both dynamic and static information about the driving scene. When any type of scene token is missing,

Table 8: Ablation on the performance under different planning manners and traffic densities.

| Planning Manner | Traffic Density | NC ↑ | DAC ↑ | TTC ↑ | Comf ↑ | EP ↑ | PDMS ↑ |
|---|---|---|---|---|---|---|---|
| Deterministic | Low | 98.1 | 97.4 | 95.2 | 99.9 | 83.2 | 89.4 |
| Probabilistic | Low | 98.3 | 99.0 | 95.5 | 100 | 83.5 | 90.6 |
| Deterministic | Medium | 97.8 | 96.3 | 94.8 | 100 | 81.9 | 87.5 |
| Probabilistic | Medium | 98.4 | 96.8 | 95.4 | 100 | 82.3 | 89.0 |
| Deterministic | High | 97.3 | 95.1 | 93.6 | 100 | 79.8 | 85.8 |
| Probabilistic | High | 98.0 | 97.4 | 94.3 | 100 | 82.0 | 87.7 |

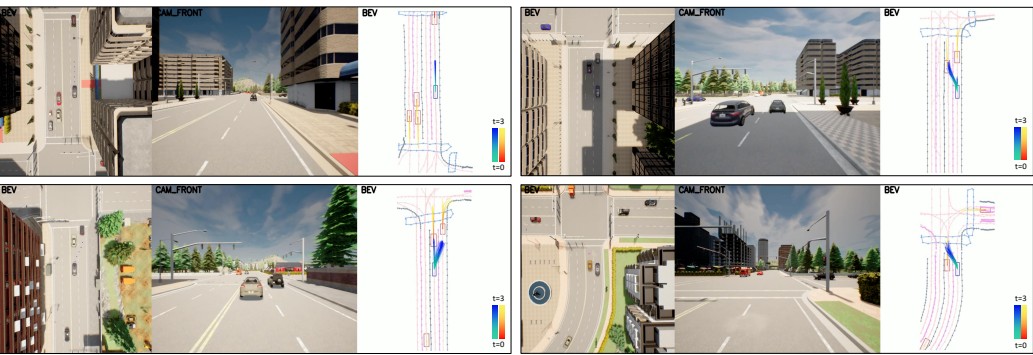

Figure 4: Qualitative results of VADv2 on the CARLA Town05 Long benchmark.

the model's planning performance will be affected (ID 3-ID 6). The best planning performance is achieved when the model incorporates all of the aforementioned designs (ID 7).

**Planning Robustness.** Based on the number of dynamic agents within 20 m of the ego vehicle, we categorize traffic density as Low (<5), Medium (5–10), and High (>10), and evaluate performance under different planning manners and density levels (Table 8). For deterministic planning, we modify the planning head of VADv2 to an MLP that directly regress future trajectories, following common practice (Jiang et al., 2023). Results show that probabilistic planning consistently outperforms its deterministic counterpart. Furthermore, while probabilistic planning maintains stable performance across varying density scenarios, deterministic planning exhibits noticeable degradation. These findings highlight the effectiveness and robustness of our probabilistic planning paradigm. Additional experiments and ablations are provided in Appendix A due to page limits.

## 4.5 VISUALIZATION

Figure 4 presents qualitative results of VADv2. The top left image shows multi-modal planning trajectories at different driving speeds. The top right shows both forward creeping and left lane-changing. The bottom left shows a right-lane change scenario at an intersection, where both temporary straight and immediate lane-change trajectories are predicted. The bottom right depicts a lane-change with a vehicle in the target lane, and VADv2 generates multiple reasonable trajectories.

## 5 CONCLUSIONS AND LIMITATIONS

In this work, we introduce VADv2, an end-to-end driving model based on probabilistic planning. It runs stably in the CARLA simulator and achieves state-of-the-art closed-loop performance, significantly outperforming existing methods. Comprehensive experiments on NAVSIM and the 3DGS-based benchmark further demonstrate its effectiveness and robustness in complex driving scenarios. The feasibility of this probabilistic planning paradigm is primarily validated.

Currently, both simulator and 3DGS-based closed-loop environments still present limitations, such as naive agent behaviors and insufficient scene realism, which may restrict the performance of VADv2. In future work, we plan to explore how large-scale expert driving data can be leveraged to further enhance planning performance and bridge the gap between simulation and real-world deployment.

## ACKNOWLEDGMENTS

This work was supported by the National Natural Science Foundation of China (NSFC No. 62276108).

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

Table 9: Closed-loop results on Town05 Short.

| Method | Modality | DS ↑ | RC ↑ |
|---|---|---|---|
| CILRS | C | 7.5 | 13.4 |
| LBC | C | 31.0 | 55.0 |
| Transfuser | C+L | 54.5 | 78.4 |
| ST-P3 | C | 55.1 | 86.7 |
| VAD | C | 64.3 | 87.3 |
| LeapVAD | C | 88.2 | 99.5 |
| VADv2 | C | 89.7 | 93.0 |

Table 10: Ablation on vocabulary size.

| Size | L2 (m) ↓ | | | Collision (%) ↓ | | |
|---|---|---|---|---|---|---|
| | 1s | 2s | 3s | 1s | 2s | 3s |
| 256 | 0.110 | 0.207 | 0.337 | 0.000 | 0.019 | 0.057 |
| 512 | 0.099 | 0.189 | 0.313 | 0.000 | 0.022 | 0.045 |
| 1024 | 0.093 | 0.175 | 0.293 | 0.000 | 0.020 | 0.044 |
| 2048 | 0.088 | 0.173 | 0.294 | 0.000 | 0.017 | 0.041 |
| 4096 | 0.082 | 0.169 | 0.290 | 0.000 | 0.010 | 0.039 |

Table 11: Ablation on training clips.

| Amount | L2 (m) ↓ | | | Collision (%) ↓ | | |
|---|---|---|---|---|---|---|
| | 1s | 2s | 3s | 1s | 2s | 3s |
| $1 \times 10^5$ | 0.121 | 0.264 | 0.461 | 0.015 | 0.061 | 0.107 |
| $3 \times 10^5$ | 0.082 | 0.169 | 0.290 | 0.000 | 0.010 | 0.039 |
| $1 \times 10^6$ | 0.073 | 0.153 | 0.267 | 0.000 | 0.008 | 0.027 |
| $3 \times 10^6$ | 0.072 | 0.133 | 0.225 | 0.000 | 0.000 | 0.007 |

Table 12: Ablation on planning manners.

| Planning | DS ↑ | RC ↑ | L2(m) @3s↓ | Collision(%) @3s↓ |
|---|---|---|---|---|
| Deterministic | 74.6 | 95.1 | 0.223 | 0.006 |
| Probabilistic | 85.1 | 98.4 | 0.225 | 0.007 |

# A  APPENDIX

## A.1  CLOSED-LOOP RESULTS ON THE TOWN05 SHORT BENCHMARK

We summarize the Town05 Short benchmark results of VADv2 in Table 9. This benchmark evaluates targeted driving behaviors, including lane changes in dense traffic and before intersections. VADv2 achieves the highest DS score, while LeapVAD exhibits higher RC but lower DS, suggesting more infractions. Overall, the results highlight the strong and reliable driving capability of VADv2 in challenging scenarios.

## A.2  MORE ABLATION STUDY

**Vocabulary Size.** We ablate about the vocabulary size in Table 10. With the vocabulary size increasing, both L2 and collision metrics become better. A larger vocabulary size can better represent the action space with less discretization error.

**Amount of Training Clips.** Table 11 is the ablation experiments about the amount of clips of driving demonstrations used for training the end-to-end model. As expected, the model achieves better L2 and collision metrics with the data amount increasing.

**Probabilistic *vs*. Deterministic.** Ablation results on the Town05 Long benchmark (Table 12) show that deterministic and probabilistic planning perform similarly in open-loop evaluation. However, in closed-loop settings, probabilistic planning achieves notably better stability and performance, while deterministic planning struggles with planning uncertainty.

**Vocabulary Sampling.** In Table 13, we report discrete errors and planning performance for different vocabulary sampling strategies. For each training sample in NAVSIM, we select the vocabulary

Table 13: Ablation on the discretization error and performance of vocabulary sampling strategies.

| Strategy | Avg. L2 ↓ | Max L2 ↓ | NC ↑ | DAC ↑ | TTC ↑ | Comf ↑ | EP ↑ | PDMS ↑ |
|---|---|---|---|---|---|---|---|---|
| k-means | 0.132 | 0.217 | 97.9 | 97.2 | 94.5 | 100 | 81.9 | 89.0 |
| K-disks | 0.128 | 0.204 | 98.1 | 97.3 | 94.8 | 100 | 82.2 | 89.1 |
| nuScenes | 0.116 | 0.212 | 98.2 | 97.1 | 95.4 | 100 | 82.5 | 89.1 |
| FTS | 0.102 | 0.181 | 98.3 | 97.6 | 95.1 | 100 | 82.3 | 89.3 |

Table 14: More detailed statistics of our real world 3DGS-based validation dataset.

Figure 5: Visualization of the planning vocabulary on the CARLA Town05 benchmark.

| Scenario | Type | Percentage |
|---|---|---|
| Sunny | Weather | 74.78% |
| Night & Rainy | Weather | 25.22% |
| Crowded Road | Precise Behavior | 6.23% |
| Narrow Road | Precise Behavior | 6.82% |
| Intersection | Precise Behavior | 38.58% |
| Cut-in | Interactive Scenario | 9.79% |
| Ped. Crossing | Interactive Scenario | 9.20% |

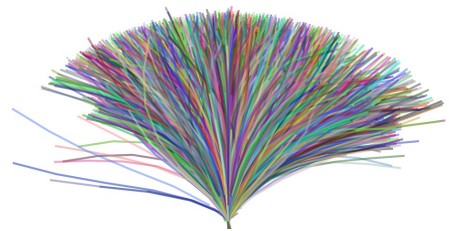

Table 15: More evaluation details on the 3DGS-based benchmark.

| Method | Reference | Parameters | Latency | Collision (%) | Platform |
|---|---|---|---|---|---|
| TransFuser | TPAMI 22 | 0.36B | 118ms | 0.320 | RTX 4090 |
| VAD | ICCV 23 | 0.36B | 118ms | 0.335 | RTX 4090 |
| GenAD | ECCV 24 | 0.38B | 121ms | 0.341 | RTX 4090 |
| VADv2 | Ours | 0.40B | 125ms | 0.270 | RTX 4090 |

trajectory that yields the minimum L2 distance and then measure the per-timestep L2 error. We derive both the Average and Max L2 Errors and average them across all samples.

The choice of strategy leads to only minor differences, with Furthest Trajectory Sampling (FTS) achieving the best action space coverage and strongest results. We also build the vocabulary by sampling trajectories from the nuScenes training set and evaluate on NAVSIM, where performance remains comparable, indicating that VADv2 can effectively generalize across scenarios.

### A.3 More evaluation details of the 3DGS-based benchmark

Table 14 showcases the diverse and challenging test scenarios in our dataset, enabling more robust evaluation. We also assess 3DGS-based reconstruction under different weather conditions, achieving PSNR (Peak Signal-to-Noise Ratio) metrics of 29.5, 28.8, and 28.2 for sunny, rainy, and nighttime scenes, respectively, demonstrating a leading level of performance.

Table 15 reports the inference latency, model parameters, and hardware platforms of baseline methods like TransFuser and VAD. To fairly compare planning performance, we use the same perception backbone across all methods. Thus, differences in latency mainly arise from the planning module design. While VADv2 adds some overhead with its planning vocabulary, its latency remains comparable to other baselines and notably improves the primary collision rate metric.

We provide additional details of our real-world dataset and compare it with popular benchmarks in Table 16. While nuScenes is dominated by straight-driving scenarios and NAVSIM features more turns but limited data, our dataset stands out in both scale and scene diversity.

### A.4 LLM Usage

We only use LLMs to check grammar and polish writing in this paper, and the authors take full responsibility for all content.

Table 16: Quantitative analysis of the real-world 3DGS-based dataset.

| Dataset | Duration | Environment | Straight | Turning |
|---|---|---|---|---|
| nuScenes | 5.55h | Urban | 92.80% | 7.20% |
| NAVSIM | 120h | Urban | 66.40% | 33.60% |
| Ours | 2000h | Urban, Highway, Rural | 52.50% | 47.50% |

