# OpenReview forum: "VADv2: End-to-End Vectorized Autonomous Driving via Probabilistic Planning"
_ICLR.cc/2026/Conference — ICLR 2026 Poster_

### Official Review · Reviewer_HtAi · 2025-10-22

**Soundness:** 2
**Presentation:** 3
**Contribution:** 2
**Rating:** 4
**Confidence:** 4

**Summary:**

The paper introduces VADv2, a probabilistic planning model for end-to-end autonomous driving. The model tokenizes planning trajectories into discretize spaces with a vocabulary and output the probabilistic distribution of action.  Distribution loss, conflict loss and scene token loss are used for training. The model achieves state-of-the-art performance on CARLA Town05, NAVSIM and 3DGS-based benchmark.

**Strengths:**

1. The motivation of probabilistic planning is clear.
2. The model achieves state-of-the-art performance on both open-loop and closed-loop benchmarks.

**Weaknesses:**

1. The planning vocabulary plays a key role in proposed probabilistic planning. However, there lacks analysis on the quality of the vocabulary to support that the discrete action space formed by the vocubulary can well represent the continuous action space in real world, including qualitative results and statistical analysis.
2. There are already several methods to construct vocubulary for trajectories in autonomous driving, such as K-disks sampling in Trajeglish [1] and uniform quantization in Motionlm [2]. However, there is not any comparation between the proposed planning vocabulary sampling method and existing methods.
3. As mentioned in 2, the idea of predicting trajectories of ego or agents in discrete space with classification loss is not new in autonomous driving. The idea from motion tasks of [1] and [2] is easy to apply to end-to-end planning. Thus, the probabilistic planning lacks significant novelty as the major contribution of the paper.
4. It is better to move the “LLM Usage” paragraph from “Experimental Settings” subsection to appendix.

[1] Trajeglish: Traffic Modeling as Next-Token Prediction (ICLR 2024)

[2] Motionlm: Multi-agent motion forecasting as language modeling (ICCV 2023)

**Questions:**

1. How about the performance of VADv2 on middle tasks, including detection, online mapping and motion prediction?
2. Similar to ego planning, the future trajectories of other agents are also of uncertainty. However, the motion prediction of VADv2 is still in continues space with regression loss. Will conducting the task in discrete space improve the perfermance?

---

> ### Author Response · Authors · 2025-11-20
> **Official Comment by Authors (1/2)**
>
> We appreciate reviewer HtAi’s feedback on our work. It is encouraging to see our motivation for probabilistic planning recognized, as well as the strong performance of VADv2 across both open-loop and closed-loop benchmarks.
>
> We also take note of your comments on the evaluation of our planning vocabulary, comparisons with existing discretization methods, and performance on intermediate tasks. We address each point in detail below.
>
> **Q1**: Further analysis on the planning vocabulary. (Weaknesses 1)
>
> **R1**: Thank you for the thoughtful comment. We will explicitly include a qualitative visualization of the vocabulary in the revised manuscript. In addition, Appendix Table 6 provides an ablation on vocabulary size. Following your suggestion and that of Reviewer bZ8f, we further evaluate the impact of different vocabulary sampling strategies (Table R1), as well as the corresponding discretization errors (Table R3). These ablations demonstrate the capability of VADv2’s vocabulary to represent the continuous action space.
>
> **Table.R1 Ablation on different vocabulary sampling strategies.**
>
> | Strategy | NC ↑ | DAC ↑ | TTC ↑ | Comf ↑ | EP ↑ | PDMS ↑ |
> |-----|-----|-----|-----|-----|-----|-----|
> | k-means     | 97.9 | 97.2 | 94.5 | 100 | 81.9 | 89.0 |
> | K-disks [1] | 98.1 | 97.3 | 94.8 | 100 | 82.2 | 89.1 |
> | FTS         | 98.3 | 97.6 | 95.1 | 100 | 82.3 | 89.3 |
>
> **Table.R3 Ablation on the discretization error of different vocabulary sampling strategies.**
>
> | Strategy | Average L2 Error | Max L2 Error | PDMS ↑ |
> |-----|-----|-----|-----|
> | k-means | 0.132 | 0.217 | 89.0 |
> | K-disks [1] | 0.128 | 0.204 | 89.1 |
> | FTS | 0.102 | 0.181 | 89.3 |
>
> **Q2**: Comparing vocabulary construction strategy with existing methods. (Weaknesses 2)
>
> **R2**: Thank you for your insightful comment. Trajeglish [1] and MotionLM [2] are valuable contributions in constructing motion vocabularies. However, rather than focusing on vocabulary sampling strategy, the main contribution of VADv2 lies in the probabilistic planning formulation, along with model architecture design and training method, which replaces deterministic planning paradigm commonly used in prior works.
>
> Below we summarize the differences in vocabulary sampling strategies between VADv2 and these methods:
>
> Trajeglish and MotionLM discretize single-step action spaces, where each token corresponds to the action at one step. In contrast, each planning token in VADv2 represents a full action trajectory. This design brings three key benefits:
> - Planning is completed in a single forward pass without iterative rollout;
> - Each planning token corresponds to an inherently feasible trajectory, whereas iterative rollout predictions may violate dynamical constraints and require extra post-processing;
> - Discretization errors may accumulate during iterative rollout and affect prediction quality;
>
> We additionally evaluate alternative sampling strategies, as shown in Table R1, and furthest trajectory sampling achieves the best overall performance.
>
> Following your suggestion, we will incorporate a more detailed discussion and comparison with existing methods into the final version of the paper.

---

> ### Author Response · Authors · 2025-11-26
> **Official Comment by Authors (2/2)**
>
> **Q3**: Novelty of the probabilistic planning idea. (Weaknesses 3)
>
> **R3**: We appreciate the reviewer’s valuable question. Although prior motion-prediction works also adopt a discrete action space with classification loss, they differ from VADv2 in two aspects:
>
> - In Trajeglish [1] and MotionLM [2], each token represents a single-step action, while each planning token in VADv2 represents a complete trajectory. The iterative prediction in [1] and [2] can produce final trajectories that violate dynamical constraints, which is acceptable in motion prediction but not in planning.
> - [1] and [2] simply map discrete actions to action embeddings based on action-token indices, VADv2 instead uses the trajectory coordinates and applies a cosine-encoding strategy to embed them into high-dimensional planning tokens. Compared with simple index-based mapping, this design brings semantically similar actions closer in the embedding space, while the high-frequency components preserve fine-grained trajectory details.
>
>
> **Q4**: Move “LLM Usage” paragraph to appendix. (Weaknesses 4)
>
> **R4**: Thank you for the suggestion. We will move “LLM Usage” paragraph to appendix in the final paper as you recommended.
>
>
> **Q5**: Performance of VADv2 on middle tasks. (Question 1)
>
> **R5**: Thank you for the thoughtful comment. We have added VADv2’s performance on the middle tasks of the NAVSIM dataset, as shown in Table R9. The perception range for detection and motion prediction is ±50m, while the online mapping range covers 60m longitudinally and 30m laterally.
>
> **Table.R9 Ablation on the perception and motion prediction performance.**
>
> | Task| Object Detection (mAP) | Online Mapping (mAP) | Motion Prediction (minADE) |
> |-----|-----|-----|-----|
> | Score | 0.417 | 53.5 | 0.783 |
>
>
> **Q6**: Conducting motion prediction in discrete space. (Question 2)
>
> **R6**:Thank you for the valuable comment. Following your suggestion, we modified the motion prediction module to use a discrete-space classification approach following VADv2. As shown in Table R10, this led to a moderate improvement in motion prediction, while planning performance remained generally consistent.
>
> **Table.R10 Ablation on different motion prediction strategies.**
>
> | Motion Prediction Strategy | Motion Prediction (minADE) | Planning (PDMS) |
> |-----|-----|-----|
> | Continuous Space with Regression | 0.783 | 89.3 |
> | Discrete Space with Classification | 0.725 | 89.2 |

---

### Official Review · Reviewer_JaXb · 2025-10-29

**Soundness:** 2
**Presentation:** 2
**Contribution:** 2
**Rating:** 4
**Confidence:** 4

**Summary:**

The paper proposes VADv2, a probabilistic planning model for end-to-end autonomous driving that discretizes the action space into planning tokens to better handle uncertainty. The method achieves state-of-the-art closed-loop results on CARLA Town05 and is further validated on the NAVSIM dataset and a large-scale 3DGS-based benchmark.

**Strengths:**

1. The paper is easy to read.

2. Modeling the uncertainty of trajectory is important.

3. The experiments are conducted on several benchmarks.

**Weaknesses:**

The main results on CARLA may be outdated; the paper should compare against state-of-the-art methods on more challenging benchmarks such as Bench2Drive [1].

On NAVSIM, the paper does not compare with the latest state-of-the-art algorithms.

Modeling trajectory uncertainty is important; however, there are potentially better approaches such as diffusion models [2,3] or flow matching models [4], which require further comparison and discussion.

[1] Jia X, Yang Z, Li Q, et al. Bench2drive: Towards multi-ability benchmarking of closed-loop end-to-end autonomous driving[J]. Advances in Neural Information Processing Systems, 2024, 37: 819-844.

[2] Liao B, Chen S, Yin H, et al. Diffusiondrive: Truncated diffusion model for end-to-end autonomous driving[C]//Proceedings of the Computer Vision and Pattern Recognition Conference. 2025: 12037-12047.

[3] Zheng Y, Liang R, Zheng K, et al. Diffusion-based planning for autonomous driving with flexible guidance[J]. arXiv preprint arXiv:2501.15564, 2025.

[4] Xing Z, Zhang X, Hu Y, et al. Goalflow: Goal-driven flow matching for multimodal trajectories generation in end-to-end autonomous driving[C]//Proceedings of the Computer Vision and Pattern Recognition Conference. 2025: 1602-1611.

**Questions:**

N/A

---

> ### Author Response · Authors · 2025-11-20
>
> We sincerely thank reviewer JaXb for the constructive feedback. We appreciate the recognition of the importance of modeling trajectory uncertainty and your acknowledgement that our paper supported by evaluations across multiple benchmarks.
>
> We also value the reviewer’s insightful suggestions regarding more comparisons and further discussion of recent probabilistic approaches. We address these points in detail below.
>
> **Q1**: Compare against SOTA methods on more challenging benchmarks. (Weaknesses 1)
>
> **R1**: We thank the reviewer for the valuable comment. Following your suggestion, we additionally reported performance on the Bench2Drive and NAVSIMv2 benchmarks. As shown in Tables R6 and R7, VADv2 achieves SOTA planning performance on both benchmarks.
>
> **Table.R6 Closed-loop results on the Bench2Drive benchmark.**
> | Method | Reference | DS ↑ | SR ↑ | Efficiency ↑ | Comf ↑ |
> |-----|-----|-----|-----|-----|-----|
> | SparseDrive | ICRA'25 | 44.54 | 16.71 | 170.21 | 48.63 |
> | MomAD | CVPR'25 | 47.91 | 18.11 | 174.91 | 51.20 |
> | DriveTransformer | ICLR'25 | 63.46 | 35.01 | 100.64 | 20.78 |
> | ETA | ICCV'25 | 74.33 | 48.33 | 186.04 | 25.77 |
> | VADv2 | Ours | 76.15 | 50.46 | 178.24 | 37.81 |
>
> **Table.R7 Performance on the NAVSIMv2 benchmark with Extended Metrics.**
> | Method | Modality | NC ↑ | DAC ↑ | DDC ↑ | TL ↑ | EP ↑ | TTC ↑ | LK ↑ | HC ↑ | EC ↑ | EPDMS ↑ |
> |----|----|----|----|----|----|----|----|----|----|----|----|
> | Transfuser | C+L | 96.9 | 89.9 | 97.8 | 99.7 | 87.1 | 95.4 | 92.7 | 98.3 | 87.2 | 76.7 |
> | HydraMDP++ | C | 97.2 | 97.5 | 99.4 | 99.6 | 83.1 | 96.5 | 94.4 | 98.2 | 70.9 | 81.4 |
> | PRIX | C | 98.0 | 95.6 | 99.5 | 99.8 | 87.4 | 97.2 | 97.1 | 98.3 | 87.6 | 84.2 |
> | VADv2 | C | 98.0 | 98.3 | 99.4 | 99.8 | 87.1 | 96.8 | 95.2 | 98.3 | 88.1 | 85.8 |
>
> **Q2**: Compare with the latest algorithms on NAVSIM. (Weaknesses 2)
>
> **R2**: Thank you for the helpful comment. We have updated our results on NAVSIM and compared with the latest SOTA algorithms in Table R8.
>
> **Table.R8 Performance on the NAVSIM navtest benchmark.**
> | Method | Reference | Modality | NC ↑ | DAC ↑ | TTC ↑ | Comf ↑ | EP ↑ | PDMS ↑ |
> |-----|-----|-----|-----|-----|-----|-----|-----|-----|
> | GoalFlow | CVPR'25 | C+L | 98.3 | 93.8 | 94.3 | 100 | 79.8 | 85.7 |
> | Hydra-MDP++ | arXiv'25 | C | 97.6 | 96.0 | 93.1 | 100 | 80.4 | 86.6 |
> | WoTE | ICCV'25 | C+L | 98.5 | 96.8 | 94.9 | 99.9 | 81.9 | 88.3 |
> | Hydra-NeXt | arXiv'25 | C+L | 98.1 | 97.7 | 94.6 | 100 | 81.8 | 88.6 |
> | VADv2 | Ours | C | 98.3 | 97.4 | 95.7 | 100 | 82.3 | 89.3 |
>
> **Q3**: Further comparison with recent approaches on modeling trajectory uncertainty. (Weaknesses 3)
>
> **R3**: We thank the reviewer for pointing out recent advances in modeling trajectory uncertainty. Below, we summarize the key differences between VADv2 and these approaches:
>
> - Diffusion Planner [3] models uncertainty with diffusion method, and jointly predicting ego and other agents’ motions. However, it depends on GT perception and HD maps instead of raw sensor data. Its multi-step denoising also introduces substantial inference latency.
> - DiffusionDrive [2] proposes a truncated diffusion model conditioned on a small set of trajectory anchors, enabling fewer denoising steps. However, using only 20 anchors may constrain the diversity and expressiveness of the final trajectories. Increasing the number of anchors could mitigate this issue, but each anchor requires its own denoising process, making inference computationally expensive.
> - GoalFlow [4] applies flow matching to achieve faster inference and higher-quality generation. It decomposes planning into goal-point selection followed by trajectory generation. However, relying on a single selected goal point provides a strong prior that can restrict trajectory diversity. In addition, GoalFlow adopts hand-crafted rules and thresholds to select the final trajectory, which limits its generality and scalability.
>
> In comparision, VADv2 offers the following advantages:
>
> - Prior methods generate trajectories directly in continuous space, which may violate vehicle dynamics and require post-processing. VADv2 instead frames planning as a classification task over the planning vocabulary, ensuring all trajectories are dynamically feasible.
> - VADv2 performs single-shot classification rather than iterative denoising of diffusion, making it more suitable for real-world application. It reaches inference speed comparable to mainstream models while delivering significantly better performance. Moreover, unlike DiffusionDrive’s small set of predefined anchors, VADv2 uses a much larger vocabulary and can flexibly add new candidate actions during inference.
> - VADv2 naturally outputs relative preference scores over all candidate trajectories, enabling principled and flexible integration of learning- or rule-based trajectory selection strategies.
>
> Thank you again for your valuable comments, we will incorporate the updated experiments and discussions into the revised paper as you suggested.

---

> ### Comment · Reviewer_JaXb · 2025-11-27
>
> Thank you for your reply. If I remember correctly, the NavSim scores appear to have been updated from the previous version. Could you kindly share what changes were made to the paper? I would really appreciate it if you could point out the specific revisions.

---

> ### Author Response · Authors · 2025-11-27
>
> Certainly, and thank you for your question. In the revised paper, we made the following updates:
>
> - We updated the VADv2 results on NavSim in Table 2 and, following your suggestion, added the latest algorithms for comparison.
> - As recommended by Reviewer HtAi, we moved the ''LLM Usage'' section to the Appendix.
>
> We have uploaded the revised PDF and highlighted the changes in blue for your convenience. We hope this clarifies the discrepancy between versions.
>
> During the review period, we re-ran the NavSim experiments using the same training protocol and hyperparameter search procedure that we applied to the other datasets. This procedure had not been fully executed for NavSim in the initial submission due to time constraints near the deadline. The updated results ensure that the NavSim evaluation meets the same level of experimental rigor as the rest of our experiments (while still allowing dataset-specific settings).
>
> These modifications affect only the numerical values in Table 2, no methodological components or conclusions of the paper were changed. All updated numbers are clearly marked in the revised PDF.
>
> We also conducted experiments on more benchmarks as you recommended, which further strengthens the comprehensiveness of the evaluation. All additional experiments and discussions will be fully included in the final version, based on the valuable feedback from you and the other reviewers.

---

### Official Review · Reviewer_biv4 · 2025-10-29

**Soundness:** 3
**Presentation:** 3
**Contribution:** 3
**Rating:** 8
**Confidence:** 5

**Summary:**

This paper presents VADv2, an end-to-end vision-based autonomous driving model that introduces a probabilistic planning paradigm to handle uncertainty and non-deterministic behavior in human driving. Instead of regressing deterministic trajectories or control commands, VADv2 models the planning policy as a scene-conditioned stochastic process, predicting a probabilistic distribution over discretized action tokens (“planning vocabulary”). The system tokenizes both the scene (through BEV-based map, agent, traffic, and image tokens) and the action space, learning the probability field of feasible maneuvers via a KL-based distribution loss, conflict regularization, and scene supervision. Most probable trajectory are selected for control during inference, supporting both multi-modal planning and flexible rule-based refinement. Extensive experiments on CARLA Town05, NAVSIM, and a large-scale 3DGS testing showcasing strong performance, particularly improving safety (collision, deviation ratios) and stability in closed-loop tests.

**Strengths:**

1. VADv2 introduces a well-motivated shift from deterministic to probabilistic planning, effectively addressing multi-modal and uncertain decision spaces, an enduring challenge in end-to-end driving.

2. The model outperforms strong baselines on CARLA, NAVSIM, and 3DGS. The added 3DGS benchmark enhances credibility in real-world-like evaluations.

3. The paper is well-organized and provides detailed architectural and experimental information, including ablations (vocabulary size, data scale, loss components) and insightful qualitative visualizations.

**Weaknesses:**

The study mainly varies vocabulary size and data scale, but omits analysis of probabilistic vs. deterministic stability across traffic densities, or uncertainty calibration (e.g., entropy of action distribution).

While multi-modal outputs are demonstrated qualitatively, quantitative uncertainty analysis or calibration plots are missing. It is important for safety-critical evaluation.

**Questions:**

1. How sensitive is performance to the number and diversity of sampled trajectories (vocabulary size N)? Is there a principled way to balance coverage vs. redundancy?

2. Have you evaluated whether the predicted probabilities are well-calibrated (e.g., reliability or ECE curves)? Does confidence correlate with actual driving success?

3. How does stochastic sampling affect stability under closed-loop evaluation?

4. The paper hints at integration between rule-based and learned planning based on probability confidence. How is this threshold determined and implemented in practice (e.g. HSD), and does it improve real-world safety?

---

> ### Author Response · Authors · 2025-11-20
>
> We sincerely thank reviewer biv4 for the constructive feedback. The acknowledgment of our probabilistic planning approach and its strong performance across benchmarks is greatly appreciated.
>
> We are also grateful for the reviewer‘s comments regarding uncertainty modeling, multi-modal behavior, and the integration of rule-based and learning-based planning. We provide further clarifications and analyses on these points below.
>
> **Q1**: Probabilistic v.s. Deterministic stability across traffic densities. (Weaknesses 1)
>
> **R1**: We thank the reviewer for the valuable comment. Following your suggestion, we conducted additional ablations. Based on the number of dynamic agents within 20m around ego vehicle, we categorized traffic density as Low (<5), Medium (5–10), and High (>10). We then evaluated performance under different traffic densities and planning manners, as shown in Table R4. Overall, the probabilistic approach consistently outperforms the deterministic one.
>
> **Table.R4 Ablation on the performance under different traffic densities and planning manners.**
>
> | Planning Manner | Traffic Density | NC ↑ | DAC ↑ | TTC ↑ | Comf ↑ | EP ↑ | PDMS ↑ |
> |-----|-----|-----|-----|-----|-----|-----|-----|
> | Determ. | Low | 98.1 | 97.4 | 95.2 | 99.9 | 83.2 | 89.4 |
> | Prob. | Low | 98.3 | 99.0 | 95.5 | 100 | 83.5 | 90.6 |
> | Determ. | Medium | 97.8 | 96.3 | 94.8 | 100 | 81.9 | 87.5 |
> | Prob. | Medium | 98.4 | 96.8 | 95.4 | 100 | 82.3 | 89.0 |
> | Determ. | High | 97.3 | 95.1 | 93.6 | 100 | 79.8 | 85.8 |
> | Prob. | High | 98.0 | 97.4 | 94.3 | 100 | 82.0 | 87.7 |
>
>
> **Q2**: Quantitative uncertainty analysis of multi-modal outputs. (Weaknesses 2)
>
> **R2**: Thank you for the valuable suggestion. We further evaluated the multi-modal outputs of VADv2. Beyond the highest-probability trajectory, we also considered other candidates within the top-5 set, as shown in Table R5. Results indicate that the top-1 trajectory achieves the best overall performance, and the performance of other candidates is basically comparable to the top-1, showing that VADv2 can generate high-quality multi-modal outputs.
>
> **Table.R5 Ablation on the performance of multi-modal outputs.**
>
> | Trajecotry | NC ↑ | DAC ↑ | TTC ↑ | Comf ↑ | EP ↑ | PDMS ↑ |
> |-----|-----|-----|-----|-----|-----|-----|
> | Top1 | 98.3 | 97.4 | 95.7 | 100 | 82.3 | 89.3 |
> | Top2 | 98.2 | 97.3 | 94.8 | 100 | 82.2 | 89.1 |
> | Top3 | 98.0 | 96.7 | 94.6 | 99.8 | 82.0 | 88.3 |
> | Top4 | 97.8 | 96.2 | 93.2 | 99.9 | 81.7 | 87.6 |
> | Top5 | 97.4 | 96.1 | 93.5 | 99.7 | 81.4 | 87.5 |
>
>
> **Q3**: More clarification on balancing vocabulary size/coverage v.s. redundancy. (Question 1)
>
> **R3**: Thank you for the valuable comment. We provide an ablation on vocabulary size in Appendix Table 6. In general, a larger vocabulary improves planning performance. Beyond a certain point (e.g., 1024), increasing the vocabulary still helps but with diminishing returns. With sufficient training data, a larger vocabulary better models the action space. During inference, the number of action candidates can be dynamically adjusted to balance coverage and redundancy.
>
>
> **Q4**: Evaluating the reliability of predicted probabilities. (Question 2)
>
> **R4**: Thank you for your insightful comment. Following your suggestion and the issue raised in Weaknesses 2, we report the planning performance of the top-5 predicted trajectories in Table R5. Even the lower-ranked candidates show reasonable reliability. In fact, VADv2’s predicted probabilities reflect a relative ranking among trajectories rather than an absolute confidence in driving success.
>
>
> **Q5**: Stability of stochastic sampling.  (Question 3)
>
> **R5**: Thank you for your insightful comment. As noted in Section 3.4 (Line 316), our closed-loop evaluation does not use stochastic sampling; instead, we select the action with the highest probability. This yields more consistent driving behavior.
>
> In real-world deployment, the predicted planning distribution can be combined with rule-based selection to prevent the model from getting stuck in local optima or to select more efficient actions (e.g., overtaking) in complex scenarios, thereby improving the system’s upper-bound performance.
>
>
> **Q6**: Integration between rule-based and learning-based planning in practice; (Question 4)
>
> **R6**: Thank you for your constructive comment. The probability predicted by VADv2 is a relative score among candidates rather than an absolute measure. In practice, we use top-k selection instead of fixed thresholds, giving priority to the top-1 trajectory. We then evaluate its feasibility using rule-based checks based on perception results, such as potential collisions. If the top-1 trajectory fails these checks, we consider other candidates in the top-k set. When none of them satisfy the constraints, we refine the top-1 trajectory through an optimization-based refinement to enforce the required constraints and obtain the final trajectory.

---

> > ### Comment · Reviewer_biv4 · 2025-11-22
> >
> > Thanks for your detailed response in resolving most of my questions. In closed-loop deployment / simulations, how could VADv2 handle consistency issues regarding: 1) sudden collapse or causal confusion in lateral/longtitudal directions; 2) spatial-temporal coherence and smoothness across frames under uncertain scenarios?

---

> > > ### Author Response · Authors · 2025-11-25
> > >
> > > Thank you for your insightful comment. We agree that the consistency issues you mentioned are critical for safe and human-like driving. In both open-loop and closed-loop evaluations of VADv2, we intentionally do not apply any consistency constraints or post-processing, so that the planning performance of the model itself can be comprehensively evaluated.
> > >
> > > We believe that sudden collapses or causal confusion primarily arise from the model failing to learn the correct causal relationships during training. This issue can be mitigated in two ways:
> > > - Explicitly incorporating key scene elements such as traffic signals to help the model learn the underlying causal structure.
> > > - Improving the diversity and balance of the training data to reduce distribution biases.
> > >
> > > During deployment, trajectory smoothness can be ensured through rule-based post-processing. For example, limiting lateral/longitudinal accelerations under safe conditions to filter out trajectories with sharp, abrupt changes. If all candidates fail to meet these constraints, an optimization-based refinement can be applied to the top-1 trajectory to meet the requirements.
> > >
> > > VADv2 can predict multimodal driving behavior under uncertain scenarios but also exhibit notable planning consistency. Once an initial decision and action is made, the rollout distribution quickly converges to the corresponding behavior mode. For instance, after committing to a left-lane change, subsequent predictions focus on executing that maneuver, differing mainly in fine-grained trajectory details. This feature is shown in the CARLA visualization video in the supplementary material.
> > >
> > > Under uncertain scenarios, enforcing trajectory coherence can lead to sub-optimal or even stuck solutions, preventing the model from selecting more reasonable behaviors. For example, when the ego vehicle should change lanes to pass a parked car but coherence constraints push it to keep going straight and stop. Therefore, in practice we only rely on post-processing to enhance safety and smoothness, rather than imposing strict spatio-temporal coherence constraints.

---

### Official Review · Reviewer_bZ8f · 2025-10-30

**Soundness:** 3
**Presentation:** 3
**Contribution:** 2
**Rating:** 6
**Confidence:** 5

**Summary:**

The paper overall proposes a probabilistic-based end-to-end planner that achieves great results on several benchmarks, including a closed-loop one. The architecture is very similar to VAD, and the highlight is the use of a discrete action space. However, the author should perform a more systematic analysis on the choice of discretization and more informative ablation studies, given the high similarity to VAD. At this point, I tend to accept this paper for its comprehensive comparison with other methods and performance but I may change my rate based on the author's responses.

**Strengths:**

1. This paper proposes a probabilistic-based end-to-end planner by using a discretized action space, which is neat.
2. The writing is clear and easy to follow with illustrations.
3. The performance is great and the comparison with other works is fair and solid with closed-loop validation.

**Weaknesses:**

1. I believe most of the performance gain comes from the use of discretized action vocabulary, the author uses furthest trajectory sampling to get the vocabulary. I'm wondering if the noise of the vocabulary can affect the result? If yes, then to what extent? I would like to see more experiments on that. Also, the pre-defined vocabulary can be seen as a prior, and it can be different across datasets. I want to know the zero-shot cross-dataset performance of your method.
2. Since the vocabulary is very important, I would like to see more discussion on different ways of discretization, and its discretization error, a discussion on how this error would affect the performance would make your work more comprehensive.
3. In terms of the loss function, since you have discretized the whole action space, why use cross-entropy instead of the KL-divergence?
4. The ablation study on Tab. 4 is really hard to give useful information. For example, I want to know whether the conflict loss is useful; I can't find the exact setting that only w/ or w/o this loss. I strongly suggest the author make this up during the rebuttal.
5. Visualize the discretized action vocabulary will make the reader better understand the ideology.

**Questions:**

see weakness

---

> ### Author Response · Authors · 2025-11-20
>
> We sincerely thank reviewer bZ8f for the thorough and constructive feedback. We appreciate your recognition of our probabilistic formulation and the strong performance across benchmarks.
>
> We also value your suggestions on the action vocabulary, training loss, and additional ablations. These points help strengthen our work, and we address them in detail below.
>
> **Q1**: Vocabulary Sampling Stratergy; Zero-shot Performance. (Weaknesses 1)
>
> **R1**: Thank you for your insightful comment. We conducted additional ablation using different sampling strategies, with results summarized in Table R1. As shown, the choice of sampling strategy leads to only minor variations in planning performance. Among them, Furthest Trajectory Sampling (FTS) provides the best action space coverage and achieves the strongest result.
>
> **Table.R1 Ablation on different vocabulary sampling strategies.**
>
> | Strategy | NC ↑ | DAC ↑ | TTC ↑ | Comf ↑ | EP ↑ | PDMS ↑ |
> |-----|-----|-----|-----|-----|-----|-----|
> | k-means     | 97.9 | 97.2 | 94.5 | 100 | 81.9 | 89.0 |
> | K-disks [1] | 98.1 | 97.3 | 94.8 | 100 | 82.2 | 89.1 |
> | FTS         | 98.3 | 97.6 | 95.1 | 100 | 82.3 | 89.3 |
>
> [1] Trajeglish: Traffic Modeling as Next-Token Prediction, ICLR 2024.
>
> Due to differences in sensor calibration across datasets, e.g., the front camera FOV is 70° in nuScenes but 60° in NAVSIM, the perception module exhibits significant domain mismatch in zero-shot settings, which ultimately impacts planning. As a result, direct zero-shot evaluation across datasets is not feasible.
>
> Instead, we build the vocabulary by sampling trajectories from the nuScenes training set and train/evaluate the model on NAVSIM. As shown in Table R2, performance remains comparable, suggesting that VADv2 has the potential to generalize across scenarios when hardware calibration is consistent.
>
> **Table.R2 Ablation on robustness of the planning vocabulary across datasets.**
>
> | Vocabulary Source | NC ↑ | DAC ↑ | TTC ↑ | Comf ↑ | EP ↑ | PDMS ↑ |
> |-----|-----|-----|-----|-----|-----|-----|
> | nuScenes | 98.2 | 97.1 | 95.4 | 100 | 82.5 | 89.1 |
> | NAVSIM | 98.3 | 97.4 | 95.7 | 100 | 82.3 | 89.3 |
>
>
> **Q2**: Vocabulary Discretization Error. (Weaknesses 2)
>
> **R2**: Thank you for the valuable comment. We further report the discretization errors and planning metrics of different vocabulary sampling strategies. For each training sample in NAVSIM, we select the vocabulary trajectory that yields the minimum L2 distance and then measure the per-timestep L2 error. We derive both the Average and Max L2 Errors and average them across all samples. The results are shown in Table R3.
>
> As shown, furthest-trajectory sampling achieves the lowest error and the best planning metric. Overall, though, the performance gap remains relatively modest.
>
> **Table.R3 Ablation on the discretization error of different vocabulary sampling strategies.**
>
> | Strategy | Average L2 Error | Max L2 Error | PDMS ↑ |
> |-----|-----|-----|-----|
> | k-means | 0.132 | 0.217 | 89.0 |
> | K-disks [1] | 0.128 | 0.204 | 89.1 |
> | FTS | 0.102 | 0.181 | 89.3 |
>
>
> **Q3**: Why using cross-entropy as loss function? (Weaknesses 3)
>
> **R3**: We appreciate the reviewer’s insightful question. Our method indeed uses the KL divergence as the underlying objective. The full KL divergence between the empirical data distribution $p_{\rm data}$ and the predicted distribution $p_{\rm pred}$ is:
>
> \begin{equation}
> D_{\rm KL}(p_{\rm data} \|\| p_{\rm pred})
> = \sum_{\mathbf{a}} p_{\rm data}(\mathbf{a}) \log \frac{p_{\rm data}(\mathbf{a})}{p_{\rm pred}(\mathbf{a})}.
> \end{equation}
>
> As explained in Lines 269–272 of the manuscript, since $p_{\rm data}$ is fixed for the dataset, the term
> $p_{\rm data}(\mathbf{a}) \log p_{\rm data}(\mathbf{a})$
> is a constant with respect to the model parameters. Therefore, minimizing KL divergence is equivalent to optimizing the cross-entropy loss:
>
> \begin{equation}
> H(p_{\rm data}, p_{\rm pred}) = -\sum_{\mathbf{a}} p_{\rm data}(\mathbf{a}) \log p_{\rm pred}(\mathbf{a}).
> \end{equation}
>
> We appreciate the reviewer’s suggestion and will revise the manuscript and the formula to make the connection between KL divergence and cross-entropy more explicit.
>
>
> **Q4**: More clarification on the ablation study in Table 4. (Weaknesses 4)
>
> **R4**: Thank you for the valuable comment. In fact, the ablation study in Table 4 (ID2–ID6) each correspond to removing a single design choice. Specifically, ID2 corresponds to the experiment that only w/o the conflict loss.
>
>
> **Q5**: Visualizing the discretized action vocabulary. (Weaknesses 5)
>
> **R5**: Thank you for the helpful suggestion. Actually, the background in Figure 2’s Planning Vocabulary module already visualizes the action vocabulary we use. In the revised manuscript, we will add an explicit vocabulary visualization as you recommended.

---

> > ### Comment · Reviewer_bZ8f · 2025-11-25
> >
> > Thanks for your explanation. Most of my questions have been answered.

---

> > > ### Author Response · Authors · 2025-11-26
> > >
> > > We appreciate Reviewer bZ8f for the positive and valuable feedback, and your acknowledgment of our response. we will incorporate the relevant experiments and discussions into the revised manuscript as you suggested.

---

### Author Response · Authors · 2025-11-29
**General Response to Reviews**

We sincerely thank the Area Chair and all reviewers for the time and effort devoted to evaluating our submission. We greatly appreciate their constructive suggestions, which have helped us further strengthen the clarity, completeness, and empirical rigor of our work.

In the initial reviews, the reviewers highlighted several strengths of our submission, including:

- Clear motivation for the proposed probabilistic planning paradigm.
- Strong performance across both open-loop and closed-loop benchmarks.
- Detailed ablations and insightful visualizations.
- Overall clarity and organization.

These points of positive recognition are very encouraging. Building on these strengths, we conducted a series of targeted revisions that substantially address the reviewers’ main concerns. The updated content in the revised manuscript is highlighted in blue. Below we summarize the main updates and analyses presented in response to the reviewers’ comments:

**1. Expanded Experiments and Analyses**

- Added evaluations on more latest benchmarks, including Bench2Drive and NAVSIMv2 (Table R6 and R7).
- Included additional ablations on vocabulary sampling strategies (Table R1 and R2), discretization error (Table R3), traffic-density robustness (Table R4), multi-modal outputs (Table R5), and the motion-prediction auxiliary task (Table R9 and R10).

**2. Enhanced Technical Discussion**

- Strengthened explanations of the loss formulation in probabilistic planning, clarifying its relation to KL divergence and cross-entropy.
- Expanded discussion and comparison of recent trajectory-generation approaches, including diffusion- and flow-matching-based planners, as well as strategies from the motion prediction field.
- Clarified design choices regarding trajectory vocabulary, embedding strategies, and the potential integration of rule-based and learning-based planning for real-world deployment.

**3. Improved Clarity and Completeness of the Manuscript**

- Added qualitative visualizations of the planning vocabulary.
- Updated tables with results of the latest algorithms and reorganized sections following reviewer recommendations.
- Refined NAVSIM results using a unified training protocol for experimental consistency across datasets (Table R8).

We truly appreciate the reviewers’ constructive feedback and will carefully incorporate their valuable suggestions into the final version of the paper.

Thank you again for your time and thoughtful consideration of our work.

---

### Meta-Review · Area_Chair_wzep · 2026-01-06

**Summary:**

Reviewer bZ8f:
-  if the noise of the vocabulary can affect the result? If yes, then to what extent? I would like to see more experiments on that ;  want to know the zero-shot cross-dataset performance of your method.
- more discussion and evaluation of different ways of discretization
- why use cross-entropy instead of the KL-divergence?
- ablation in Tab. 4 hard to interpret
- visualizing the action vocabulary

Reviewer biv4
- most concerns resolved (https://openreview.net/forum?id=0a4dA6eUHN&noteId=V2tKTMs0Oo)
- how could VADv2 handle consistency issues regarding: 1) sudden collapse or causal confusion in lateral/longtitudal directions; 2) spatial-temporal coherence and smoothness across frames under uncertain scenarios?

Reviewer JaXb:
- The main results on CARLA may be outdated; the paper should compare against state-of-the-art methods on more challenging benchmarks such as Bench2Drive [1].
- On NAVSIM, the paper does not compare with the latest state-of-the-art algorithms.
- Modeling trajectory uncertainty is important; however, there are potentially better approaches such as diffusion models or flow models

Reviewer HtAi:
- lack of analysis of vocab. quality
- no comparison of vocab. construction methods
- probabilistic planning lacks novelty

**Reviewer Concerns:**

Outstanding concerns:
- Novelty of the probabilistic planning idea: the justification given describes somewhat incremental difference, despite the claimed contribution (the first contribution stated in the paper) as probabilistic planning being proposed by the paper. If the paper is accepted, the paper should be revised to tone this claim (qualify it so it's more accurate)

**Reviewer Scores:**

Reviewer bZ8f: 6->6
Reviewer biv4: 8->8
Reviewer JaXb: 4->4 or 6
Reviewer HtAi: 4->4 or 6

---

### Decision · Program_Chairs · 2026-01-26

Accept (Poster)